

# Identification and expression analysis of the DREB transcription factor family in pineapple (*Ananas comosus* (L.) Merr.)

Mengnan Chai[1,*], Han Cheng[1,*], Maokai Yan[2], SVGN Priyadarshani[1], Man Zhang[1], Qing He[1], Youmei Huang[1], Fangqian Chen[1], Liping Liu[3], Xiaoyi Huang[3], Linyi Lai[1], Huihuang Chen[1], Hanyang Cai[1] and Yuan Qin[1,2,3]

[1] State Key Lab of Ecological Pest Control for Fujian and Taiwan Crops; Key Lab of Genetics, Breeding and Multiple Utilization of Crops, Ministry of Education; Fujian Provincial Key Lab of Haixia Applied Plant Systems Biology, Fujian Agriculture and Forestry University, Fuzhou, Fujian Province, China
[2] State Key Laboratory for Conservation and Utilization of Subtropical Agro-Bioresources, Guangxi Key Lab of Sugarcane Biology, College of Agriculture, Guangxi University, Nanning, Guangxi Province, China
[3] College of Life Sciences, Fujian Agriculture and Forestry University, Fuzhou, Fujian Province, China
* These authors contributed equally to this work.

Corresponding authors
Hanyang Cai, 907591658@qq.com
Yuan Qin, yuanqin@fafu.edu.cn

## ABSTRACT

**Background:** Dehydration responsive element-binding (DREB) transcription factors play a crucial role in plant growth, development and stress responses. Although *DREB* genes have been characterized in many plant species, genome-wide identification of the *DREB* gene family has not yet been reported in pineapple (*Ananas comosus* (L.) Merr.).

**Results:** Using comprehensive genome-wide screening, we identified 20 *AcoDREB* genes on 14 chromosomes. These were categorized into five subgroups. *AcoDREBs* within a group had similar gene structures and domain compositions. Using gene structure analysis, we showed that most *AcoDREB* genes (75%) lacked introns, and that the promoter regions of all 20 *AcoDREB* genes had at least one stress response-related *cis*-element. We identified four genes with high expression levels and six genes with low expression levels in all analyzed tissues. We detected expression changes under abiotic stress for eight selected *AcoDREB* genes.

**Conclusions:** This report presents the first genome-wide analysis of the DREB transcription factor family in pineapple. Our results provide preliminary data for future functional analysis of *AcoDREB* genes in pineapple, and useful information for developing new pineapple varieties with key agronomic traits such as stress tolerance.

## INTRODUCTION

Abiotic stress, such as salinity, drought, and high or low temperatures, severely affects the growth and development of plants. To adapt to these stressors, plants have evolved complex signal transduction pathways and response mechanisms that are induced by

specific functional and regulatory proteins. Functional proteins that respond to stress include membrane proteins (transporters and water channel proteins), osmolyte biosynthesis enzymes (to produce proline, betaine, soluble sugars, etc.), detoxification enzymes (catalase, superoxide dismutase, ascorbate peroxidase, glutathione *S*-transferase, etc.), and other proteins that help protect macromolecules (LEA protein, osmotin, antifreeze proteins, mRNA binding protein, etc.). Regulatory proteins that respond to stress include transcription factors (bZIP, MYC, MYB, DREB, etc.), protein kinases (receptor protein kinase, MAP kinase, CDP kinase, transcription-regulation protein kinase, etc.), and proteinases (phospholipase C, phosphoesterases, etc.) (*Agarwal et al., 2006*). Among the regulatory proteins, transcription factors (TFs) play pivotal roles in abiotic stress responses. Specifically, they activate or repress the expression of stress-response genes by recognizing and binding to *cis*-elements in the promoters of their targets (*Golldack, Luking & Yang, 2011*; *Malhotra & Sowdhamini, 2014*; *Agarwal et al., 2017*). They are the main targets of genetic engineering for enhancing stress tolerance in crop plants (*Century, Reuber & Ratcliffe, 2008*).

Dehydration responsive element-binding (DREB) TFs enhance plant tolerance to abiotic stresses by specifically binding dehydration response element/C-repeat (DRE/CRT) *cis*-elements to control downstream gene expression. The DREB TF family belongs to the APETALA2/ETHYLENE-RESPONSIVE FACTOR (AP2/ERF) superfamily of TFs. The AP2/ERF superfamily is characterized by the AP2 domain, which is 60–70 amino acids long, and contains two conserved sequence blocks, the YRG element and the RAYD element. The YRG element is 19–22 amino acids long and contains the conserved YRG motif, which may confer DNA-binding specificity to the AP2 protein. The RAYD element has a conserved core region that can form an amphipathic α-helix in the AP2 domain (*Okamuro et al., 1997*). The AP2 domain of the DREB subfamily differs at specific amino acid sites from that of other subfamilies. These include the valine (Val14) and glutamine (Glu19) residues, which are conserved in the DREB subfamily (*Sakuma et al., 2002*).

The DREB subfamily members in *Arabidopsis thaliana* can be classified into six groups, named A-1 to A-6, or DREB1 to DREB6 (*Sakuma et al., 2002*). Of these, the TFs belonging to A-1 and A-2 are functionally well characterized. The first identified *DREB* gene was the A-1 member *AtCBF1*, which is strongly induced by low temperature. In addition, *AtDREB1A* and *AtDREB1C* positively regulate low-temperature stress responses (*Jaglo-Ottosen et al., 1998*; *Liu et al., 1998*). *SwDREB1* from sweet potato (*Ipomoea batatas*) is involved in the response to low temperature (*Kim et al., 2008*). Heterologous overexpression of zoysia grass (*Zoysia japonica*) *ZjDREB1.4* in *Arabidopsis* enhanced tolerance to high and freezing temperature stresses without obvious growth inhibition (*Feng et al., 2019*). In rice (*Oryza sativa*), the interaction of OsDREB1A, OsDREB1B and OsDREB1C with the GCC box enhanced the cold tolerance of the plants (*Donde et al., 2019*). Thus, DREB1 TFs are mainly associated with cold stress regulation.

By contrast, DREB2 is mainly associated with drought and salinity tolerance (*Liu et al., 1998*). *AtDREB2A* and *AtDREB2B*, the first reported A-2 members, are induced by dehydration and salinity (*Sakuma et al., 2002*). Overexpression of soybean (*Glycine max*) *GmDREB2* in *Arabidopsis* enhanced salinity tolerance without growth retardation

(*Chen et al., 2007*). In sugarcane (*Saccharum* spp. Hybrid), heterologous overexpression of *EaDREB2* enhanced the tolerance of plants to drought and salinity stress (*Augustine et al., 2015*).

In contrast to A-1 and A-2 proteins, the functions of A-3 to A-6 members are only beginning to be uncovered. The maize (*Zea mays*) A-4 subgroup gene *ZmDREB4.1* was associated with the negative regulation of plant growth and development (*Li et al., 2018*). A novel A-5 subgroup gene from desert moss (*Syntrichia caninervis*), *ScDREB8*, enhanced the salt tolerance of *Arabidopsis* seedlings by up-regulating the expression of stress-related genes (*Liang et al., 2017*). *CmDREB6* belongs to the A-6 subgroup, and its overexpression enhanced the tolerance of chrysanthemum (*Chrysanthemum morifolium*) to heat stress (*Du et al., 2018*).

Pineapple (*Ananas comosus* (L.) Merr.), the third most important tropical fruit in world production, is widely grown in tropical and subtropical regions (*Moyle et al., 2005*). The crop has high economic value, and pineapple cultivation is of great significance to the development of local agriculture. However, the changes in global climate have underscored how different abiotic and biotic stresses critically affect the growth of pineapple (*Mittler, 2006*; *Ray et al., 2013*). Pineapples are damaged under severe drought and high temperature. Low temperatures diminish growth. Biotic stressors such as pests, diseases, and weeds also lead to significant yield loss (*Lobo & Paull, 2016*).

Dehydration responsive element-binding family genes have been identified in *Arabidopsis thaliana* (*Hwang et al., 2012*), perennial ryegrass (*Xiong & Fei, 2006*), *Triticum* L. (*Mondini, Nachit & Pagnotta, 2015*), *Dendranthema* (*Yang et al., 2009*), *Zea mays* (*Qin et al., 2007*) and *Oryza sativa* L. (*Cui et al., 2011*; *Gumi et al., 2018*; *Matsukura et al., 2010*). According to previous research in several plant species, most *DREB* genes respond to various stress conditions. However, *DREB* genes have never been reported in pineapple. Therefore, our analysis focused on the identification of *AcoDREB* genes as well as the characteristics of the encoded DREB TFs. In this study, we identified 20 *AcoDREB* genes belonging to five subgroups and analyzed their gene and protein structures, protein motifs, chromosomal distribution and expression profiles. Our results provide a relatively complete profile of the pineapple *DREB* gene family. This may aid further functional analysis of each member, and facilitate the improvement of pineapple varieties via gene-transfer techniques, to confer tolerance to abiotic and biotic stresses (*Priyadarshani et al., 2019*).

## MATERIALS AND METHODS

### Identification of DREB family members in pineapple

Dehydration responsive element-binding amino acid sequences from *Oryza sativa* and *Arabidopsis thaliana* were obtained from the Rice Genome Annotation Project (RGAP, http://rice.plantbiology.msu.edu/index.shtml) (*Kawahara et al., 2013*) and The *Arabidopsis* Information Resource (TAIR, http://www.arabidopsis.org) (*Berardini et al., 2015*), respectively. The DREB sequences from *Arabidopsis* were used as search queries in BLAST-P against the pineapple genome. The AP2 (PF00847) domain was downloaded and used as a query to perform a HMMER search with default parameters (https://www.ebi.ac.uk/Tools/hmmer/search/phmmer). HMMER is a software package

that uses profile hidden Markov Models to identify conserved domains (*Ming et al., 2015*). Redundant sequences were eliminated and the Simple Modular Architecture Research Tool (SMART, http://smart.embl-heidelberg.de/) (*Letunic & Bork, 2018*) was used to verify the existence and completeness of the core domain within the identified sequences. The sequences that met these criteria were used for phylogenetic analysis.

## Protein characteristics and chromosomal localization

For each of the putative *AcoDREB* genes, the gene length, amino acid number, coding sequence (CDS) length, and chromosome position were collected from the Pineapple Genomics Database (PGD, http://pineapple.angiosperms.org/pineapple/html/index.html) (*Xu et al., 2018*). The molecular weights and isoelectric points of the putative proteins were predicted using the ExPASy proteomics server (http://expasy.org/) (*Gasteiger et al., 2003*). Based on the start positions of the genes and the lengths of the corresponding chromosomes, MapChart (*Voorrips, 2002*) was used to visualize the 20 *AcoDREB* genes that were mapped onto the 25 pineapple chromosomes and scaffold sequences.

## *Cis*-element analysis of *AcoDREB* gene promoters

The 2 kb upstream sequences of the *AcoDREB* genes were retrieved from the Pineapple Genomics Database and submitted to Plant *Cis*-Acting Regulatory Element (PlantCARE, http://bioinformatics.psb.ugent.be/webtools/plantcare/html/) (*Lescot et al., 2002*) to detect the presence of the following six regulatory elements (*Sazegari, Niazi & Ahmadi, 2015*): abscisic acid (ABA)-responsive elements (ABREs; ACGTG/TC), which are involved in ABA responsiveness (*Yamaguchi-Shinozaki & Shinozaki, 1993*); dehydration-responsive elements (DREs; A/GCCGAC), which are involved in plant responses to dehydration, low temperature, and salt stress (*Narusaka et al., 2003*); low temperature-responsive elements (LTREs; CCGAA), which are involved in low-temperature responses (*Roy Choudhury et al., 2008*); TC-rich repeats (G/ATTCTCT), which are involved in defense and stress responses (*Diaz-De-Leon, Klotz & Lagrimini, 1993*); W-boxes (TGACC/T), which are the binding site of WRKY TFs in defense responses (*Jiang et al., 2017*); and MBS (TAACTG), or MYB binding sites, which are involved in drought-inducibility (*Urao et al., 1993*).

## Sequence alignment and phylogenetic analysis

The CDS of the *AcoDREB* genes were obtained from the Pineapple Genomics Database and imported into DNAMAN Version 9 for sequence alignment (*Wang, 2016*). The phylogenetic tree was constructed in IQ tree using the maximum likelihood method (*Chernomor, Von Haeseler & Minh, 2016*; *Nguyen et al., 2015*). For this analysis, the parameters were set to default, except for the ultrafast bootstrap option, which was set to $n = 1,000$ (*Hoang et al., 2018*), after performing multiple sequence alignments in MUSCLE 3.7 (*Edgar, 2004*) using default parameters. To validate the maximum likelihood results, the neighbor-joining method was used to construct a tree using MEGA7 (*Kumar, Stecher & Tamura, 2016*).

## Gene structure analysis and conserved motif identification

The *DREB* gene structures, including the numbers and positions of exons and introns, were determined using the Gene Structure Display Server (GSDS, http://gsds.cbi.pku.edu.cn/) (*Guo et al., 2007*). Multiple EM for Motif Elicitation (MEME, http://meme-suite.org/tools/meme) was used to analyze the amino acid sequences of the 20 AcoDREBs; the maximum number of motifs was set to 10, and default parameters were used (*Bailey et al., 2009*).

## Plant material and growth conditions

The pineapple (*Ananas comosus*) variety MD2 was provided by the Qin Lab (Haixia Institute of Science and Technology, Fujian Agriculture and Forestry University, Fujian, China) (*Priyadarshani et al., 2018*). Plants were grown on a soil mixture containing 2:1 (v/v) peat moss:perlite, in plastic pots in a greenhouse under the following conditions: 30 °C, 60–70 µmol photons $m^{-1}$ $s^{-1}$ light intensity, 70% humidity, and a 16-h light/8-h dark photoperiod.

## RNA-Seq of different pineapple tissues

We used an RNA extraction kit (Omega Bio-Tek, Shanghai, China) to extract total RNA from the following tissues: calyx, gynoecium, ovule, petal and stamen. The tissues were collected according to previously described methods (*Chen et al., 2017*). The NEBNext Ultra RNA Library Prep Kit for Illumina was used to prepare libraries prior to sequencing. RNA-Seq data for root, leaf, leaf base, leaf tip, flower and fruit at different development stages were collected from the Pineapple Genomics Database (*Ming et al., 2015*). Using TopHat v2.1.1 (*Trapnell et al., 2012*) with default parameters, the trimmed paired-end reads of all samples were aligned to the pineapple genome. Cufflinks v2.2.1 and Cuffdiff v2.2.1 were used to estimate the Fragments Per Kilobase of exon model per Million mapped values. The heatmap showing the *AcoDREB* gene expression profile was generated using the pheatmap package in R (*Galili et al., 2018*).

## Stress treatments

One-month-old plants in rooting medium were used as the planting material for the stress treatment analyses. Uniform tissue-cultured seedlings were obtained from the Qin Lab (*Priyadarshani et al., 2018*). Seedlings were subjected to the following stress treatments: low temperature (4 °C), high temperature (45 °C), drought (350 mM mannitol), and high salt (150 mM NaCl). Root and leaf tissues were collected at 6, 12, 24 and 48 h after treatment. Seedlings that were not subjected to any of the stress treatments were used as controls. The collected samples were immediately stored in liquid nitrogen prior to total RNA extraction (*Rahman et al., 2017*).

## Quantitative real-time PCR and data analysis

Total RNA was extracted using the Plant RNA Kit (Omega Bio-Tek, Shanghai, China) according to the manufacturer's instructions. The RNA concentrations ranged from 100 to 500 ng/µl, and the $OD_{260}/OD_{280}$ ratios ranged from 1.8 to 2.0. According to the supplier's instructions for AMV reverse transcriptase (Takara Bio, Beijing, China), 1 µg of
purified total RNA was reverse transcribed into cDNA in a total reaction volume of 20 µl (*Cai et al., 2019*). To quantify the relative transcript levels of selected *DREB* genes, real-time PCR was performed with gene-specific primers on the Bio-Rad Real-time PCR system (Foster City, CA, USA) according to the manufacturer's instructions. The gene-specific primers used for this analysis are listed in Table S1. The PCR program used the following conditions: 95 °C for 30 s, 40 cycles of 95 °C for 5 s and 60 °C for 34 s and 95 °C for 15 s. For all tested genes, three technical replicates and at least three independent biological replicates were used (*Cai et al., 2017*; *Zhang et al., 2018*). Relative expression was calculated using the $2^{-\Delta\Delta Ct}$ method (*Century, Reuber & Ratcliffe, 2008*). Data were analyzed using one-way analysis of variance (ANOVA). Significant differences between treatments and controls are indicated by asterisks (* indicates a *p*-value < 0.05 and ** indicates a *p*-value < 0.01) (Table S2).

# RESULTS

## Genome-wide identification and chromosomal locations of pineapple *DREB* genes

Using *Arabidopsis* DREB amino acid sequences as search queries in BLAST, 20 DREB amino acid sequences were obtained from the pineapple proteome. The corresponding genes were named *AcoDREB1* to *AcoDREB20* (Table S3), and the amino acid sequences are listed in Table S4. Table 1 lists the following information for the 20 genes: gene name, gene ID, nucleotide and amino acid lengths, and the predicted isoelectric point (pI) and molecular weight (Mw) of the encoded protein. The protein lengths ranged from 149 (*AcoDREB13*) to 463 (*AcoDREB20*) amino acids, and the CDS lengths ranged from 450 (*AcoDREB13*) to 1392 (*AcoDREB20*) bp. The predicted protein molecular weights ranged from 16316.44 (*AcoDREB13*) to 49311.65 (*AcoDREB20*) Da, and the predicted isoelectric points ranged from 4.71 (*AcoDREB10*) to 9.68 (*AcoDREB07*) (Table S5). The 20 *AcoDREB* genes mapped to 14 pineapple chromosomes (Fig. 1), with three genes on Chr2 and two genes each on Chr3, Chr5, Chr6 and Chr17. Nine other chromosomes each contained one *AcoDREB* gene.

## Multiple sequence alignment and phylogenetic analysis of the DREB family

Multiple sequence alignment of the AcoDREB AP2 domains indicated that the domain was highly conserved among the 20 AcoDREBs, and that it displayed characteristics typical of other DREB proteins (Fig. 2). Beyond the conserved YRG and RAYD motifs, all 20 AP2 domain sequences contained a Val residue at position 14 (Val14), and 11 of them had a Glu residue at position 19 (Glu19). Val14 is more important than Glu19 for the binding of DREB to the DRE *cis*-acting elements (*Sakuma et al., 2002*).

To determine the phylogenetic relationships between the DREB family members, we constructed a multi-species phylogenetic tree using the full-length amino acid sequences of DREBs from pineapple, *Arabidopsis* (Table S6) and rice (Table S7). In Fig. 3, *AT3G57600* and *AT2G40220* (red frame) belong to the *Arabidopsis* subgroups A-2 and A-3, respectively. Because none of the pineapple *DREB* genes were homologous to the A-3

**Table 1 The *DREB* gene family in pineapple.**

| Gene ID | Gene Name | Chromosomal localization | | Amino acids length (aa) | Gene length (bp) | CDS length (bp) | Isoelectric points (pI) | Molecular weights (Mw) |
|---------|-----------|--------------------------|---|-------------------------|------------------|-----------------|-------------------------|------------------------|
| Aco000059 | AcoDREB04 | LG12:5065638-5067899 | 12 | 315 | 2,262 | 948 | 4.91 | 33,745.45 |
| Aco001190 | AcoDREB16 | LG02:13530546-13531451 | 2 | 301 | 906 | 906 | 5.66 | 33,079.93 |
| Aco001600 | AcoDREB05 | LG18:9400576-9404316 | 18 | 341 | 3,741 | 1,026 | 5.05 | 38,147.27 |
| Aco002673 | AcoDREB11 | LG06:10539056-10539706 | 6 | 216 | 651 | 651 | 5.22 | 22,927.24 |
| Aco002824 | AcoDREB17 | LG06:11885237-11886334 | 6 | 365 | 1,098 | 1,098 | 5.63 | 38,918.03 |
| Aco003376 | AcoDREB12 | LG17:2435249-2435743 | 17 | 164 | 495 | 495 | 5.79 | 18,210.66 |
| Aco006004 | AcoDREB07 | LG16:9780663-9781136 | 16 | 157 | 474 | 474 | 9.68 | 16,405.64 |
| Aco007650 | AcoDREB18 | LG08:962022-963979 | 8 | 373 | 1,958 | 1,122 | 9.07 | 40,044.4 |
| Aco008968 | AcoDREB01 | LG09:12532806-12533489 | 9 | 227 | 684 | 684 | 6.9 | 24,126.78 |
| Aco009985 | AcoDREB08 | LG10:1992629-1993102 | 10 | 157 | 474 | 474 | 9.68 | 16,405.64 |
| Aco010173 | AcoDREB06 | LG25:3102765-3103427 | 25 | 220 | 663 | 663 | 5.24 | 24,212.82 |
| Aco012243 | AcoDREB13 | LG02:73387-74171 | 2 | 149 | 785 | 450 | 9.63 | 16,316.44 |
| Aco012835 | AcoDREB09 | LG03:15051238-15052266 | 3 | 342 | 1,029 | 1,029 | 8.68 | 36,712.72 |
| Aco014268 | AcoDREB19 | LG05:128578-129975 | 5 | 221 | 1398 | 666 | 8.56 | 24,115.21 |
| Aco015162 | AcoDREB10 | LG05:1705173-1705958 | 5 | 261 | 786 | 786 | 4.71 | 27,636.53 |
| Aco016346 | AcoDREB20 | LG03:10461754-10463145 | 3 | 463 | 1,392 | 1,392 | 5.56 | 49,311.65 |
| Aco016696 | AcoDREB02 | LG17:191641-192357 | 17 | 238 | 717 | 717 | 7.66 | 26,104.49 |
| Aco018023 | AcoDREB14 | LG01:20359723-20360244 | 1 | 173 | 522 | 522 | 5.81 | 19,023.86 |
| Aco018980 | AcoDREB15 | LG02:10499315-10499860 | 2 | 181 | 546 | 546 | 9.65 | 19,006.18 |
| Aco022517 | AcoDREB03 | LG22:6333171-6333920 | 22 | 249 | 750 | 750 | 4.98 | 25,951.31 |

subgroup, we divided the *AcoDREBs* into five subgroups, I to V (Fig. 3). Group I included *AcoDREB01*, *02* and *03*, group II included *AcoDREB04*, *05*, *06* and *19*, group III included *AcoDREB07*, *08*, *09* and *10*, group IV included *AcoDREB11*, *12*, *13*, *14* and *15*, and group V included *AcoDREB16*, *17*, *18* and *20*.

## Stress-related *cis*-elements in *AcoDREB* promoters

Because of the potential involvement of *AcoDREB* genes in stress responses, we investigated the distribution of stress-related conserved *cis*-elements in their promoter regions (2 kb region upstream of the transcription start site) using PlantCARE (Table S8). The data for six abiotic stress response elements, ABRE, DRE, LTRE, TC-rich repeat, MBS and W-box, are shown in Fig. 4. All of the *AcoDREB* genes possessed at least one kind of *cis*-acting regulatory element, indicating that *AcoDREB* expression is associated with abiotic stress. Nine *AcoDREBs* had one or more LTREs, which are associated with the response to low-temperature conditions. Sixteen *AcoDREBs* contained between one and eight ABA-responsive elements, and only *AcoDREB09*, *12* and *17* had the TC-rich repeat element. Seven *AcoDREBs* had the MBS element, while W-boxes and DREs both occurred in ten *AcoDREBs*. Overall, the results of the *cis*-element analysis indicate that *AcoDREB* genes can respond to different kinds of abiotic stresses.

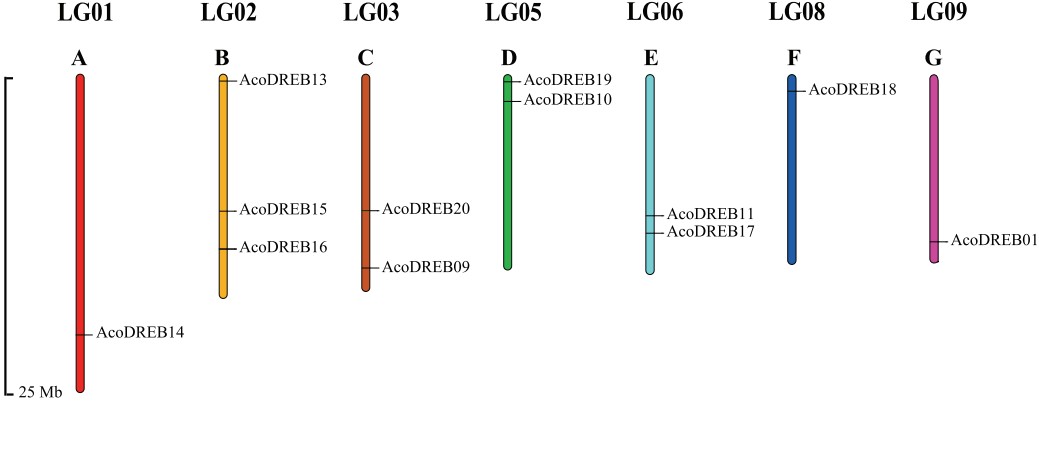

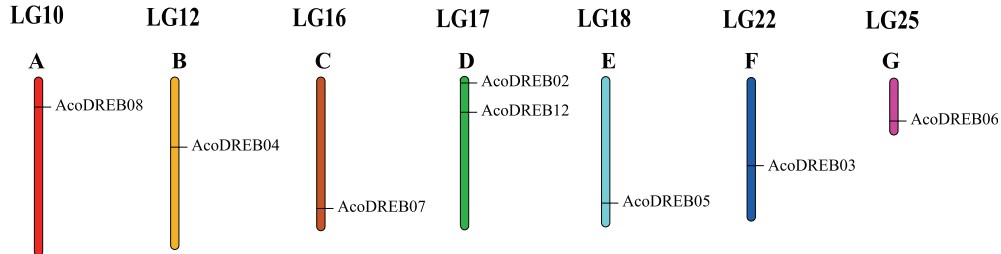

**Figure 1 Locations of *AcoDREB* genes on the pineapple chromosomes.** (A–G) Different chromosomes. The chromosome number is indicated above each bar and the length of the bar represents the size of the chromosome in pineapple. Gene star point is shown on chromosome. The figure was generated using MapChart.

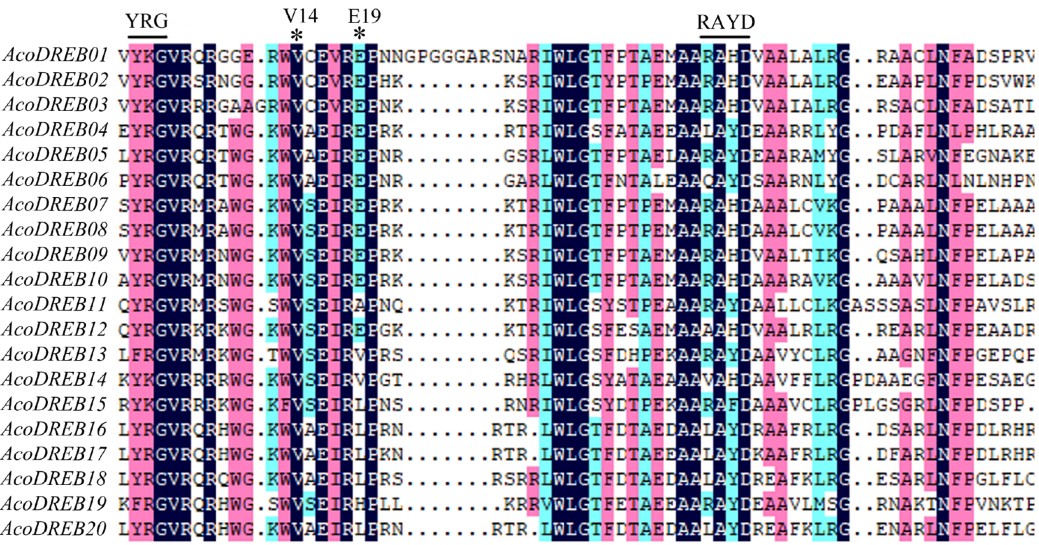

**Figure 2 Multiple sequence alignment of the AP2 domain of AcoDREB proteins.** The alignment was performed using the DNAMAN. Conserved V14, E19, YRG and RAYD motifs are highlighted by the asterisks and lines.

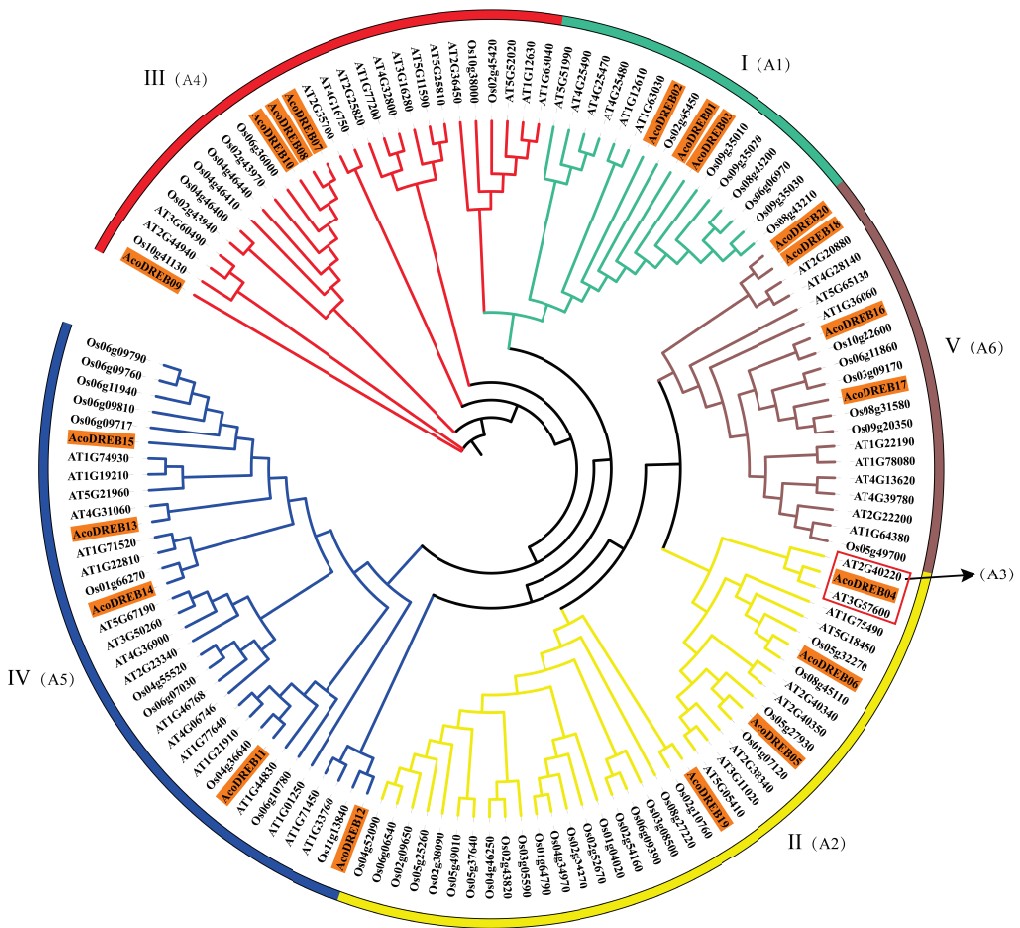

**Figure 3 Phylogenetic analysis of DREB proteins in pineapple (Aco), *Arabidopsis* and rice.** The proteins are classified into five groups: I, II, III, IV and ⬚. Classification of *Arabidopsis* by *Sakuma et al. (2002)* is indicated in parentheses. Different individual subfamilies were shown by different colors.

## *AcoDREB* gene structure and conserved motifs in the encoded proteins

Structural diversity is very common among duplicated genes, and may result in the evolution of functionally distinct paralogs. To analyze the *AcoDREB* gene structures, exon and intron numbers and positions were determined by comparing the full-length cDNA sequences to the corresponding genomic DNA sequences (Fig. 5). Seventy five percent of the *AcoDREB* genes (15/20) lacked introns. Four genes (*AcoDREB18*, *04*, *19* and *13*) had one intron each, and *AcoDREB05* had three introns. Interestingly, the members of group II differed in terms of exon and intron number as well as UTR length, which suggests that these four paralogs may have different roles in pineapple growth and development.

As shown in Fig. 6, the distribution of the motifs among AcoDREB proteins was relatively conserved. Motifs 1, 2 and 3 were present in all genes, but the motifs in different subgroups indicated some degree of divergence among them. For example, the three

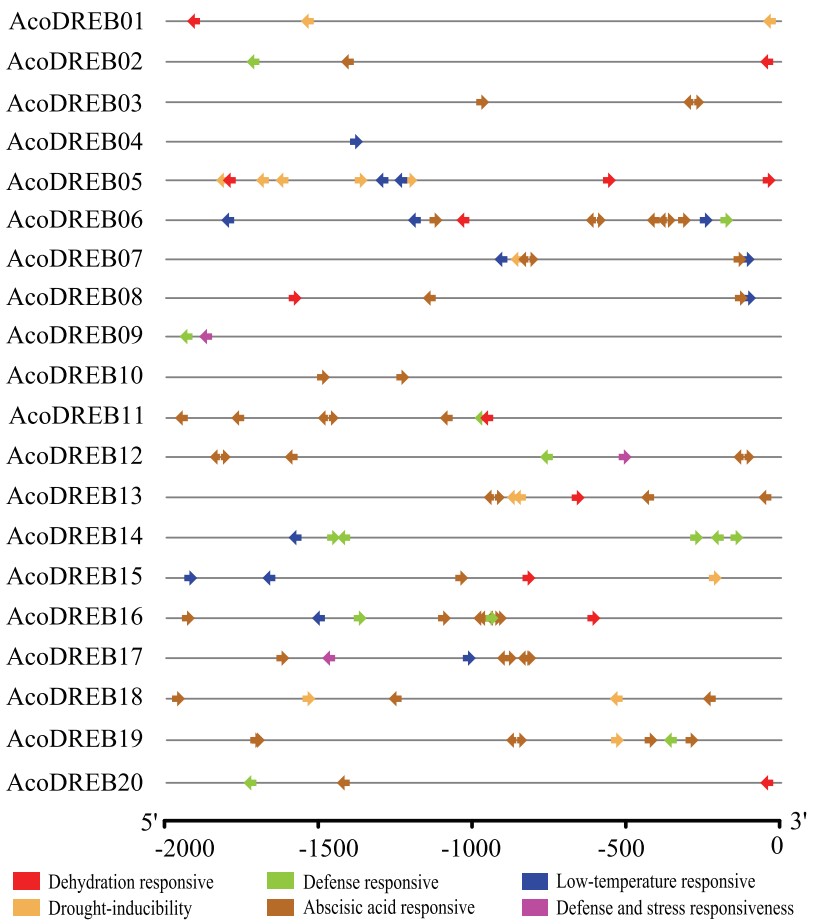

**Figure 4  Predicted *cis*-elements in *AcoDREB* promoter regions.** Promoter sequences (−2000 bp) of 20 *AcoDREB* were analyzed by PlantCARE. The upstream length to the translation start site can be inferred according to the scale.

members in subgroup I contained motifs 4, 5 and 9 in addition to motifs 1, 2 and 3. Motif 7 was only present in two of the subgroup III proteins (AcoDREB07 and AcoDREB08), and motif 4 was only present in AcoDREB05 of subgroup II. Generally, members within the same subgroup had similar motif compositions, indicating that they may perform similar functions (Fig. S1).

## *AcoDREB* gene expression profiles in different tissues at different developmental stages

The different stages of the reproductive organs were defined according to previous studies (*Azam et al., 2018*; *Su et al., 2017*). We used transcriptome sequencing data to analyze the expression patterns of the 20 *AcoDREB* genes in nine different tissues: root, leaf, flower, fruit, gynoecium, stamen, petal, calyx and ovule (Fig. 7; Table S9). We also used quantitative real-time PCR (qRT-PCR) to verify the results of the RNA-seq. All *AcoDREB* genes, except four that had low levels of expression (*AcoDREB04*, *07*, *08* and *13*), were selected for qRT-PCR analysis in seven tissues. The results obtained were consistent with the RNA-Seq expression data of these genes (Fig. 8; Table S10).

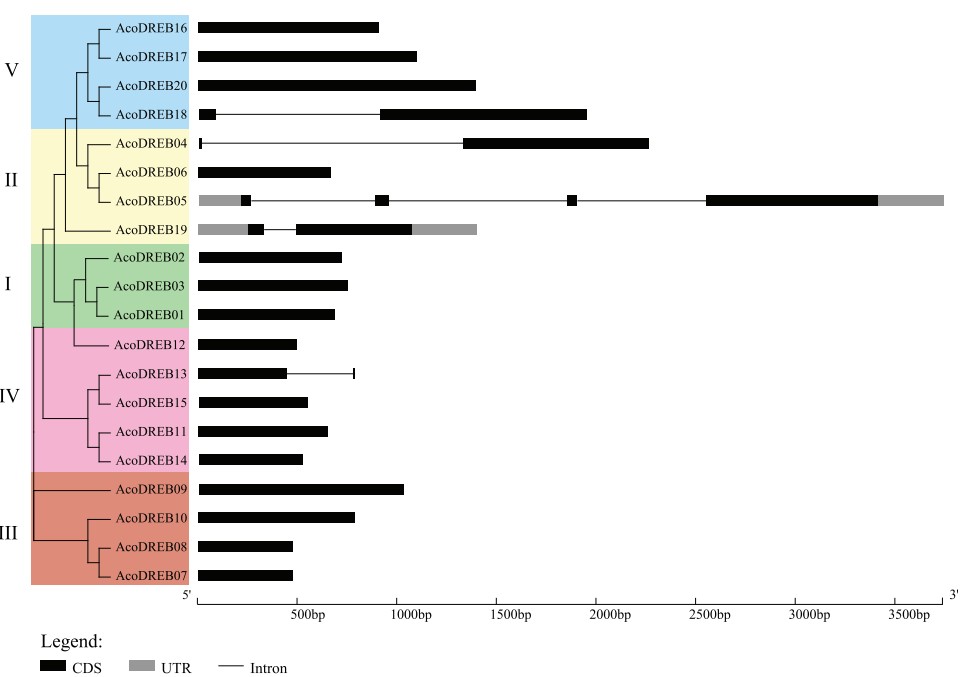

**Figure 5 Exon–intron organization of *AcoDREB* genes.** Black bars indicates exon (CDS), Gray bars indicated UTR while plain lines showing introns.

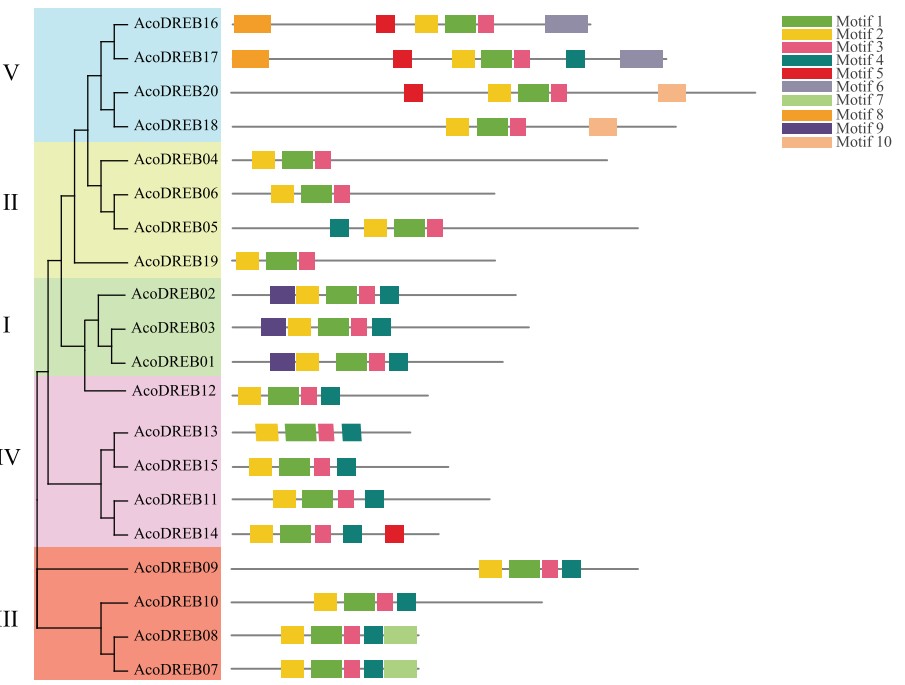

**Figure 6 The conserved motifs of the predicted AcoDREB proteins.** The conserved motifs in the AcoDREB proteins were identified with MEME software. Grey lines denote the non-conserved sequences, and each motif is indicated by a colored box. The length of motifs in each protein was presented proportionally.
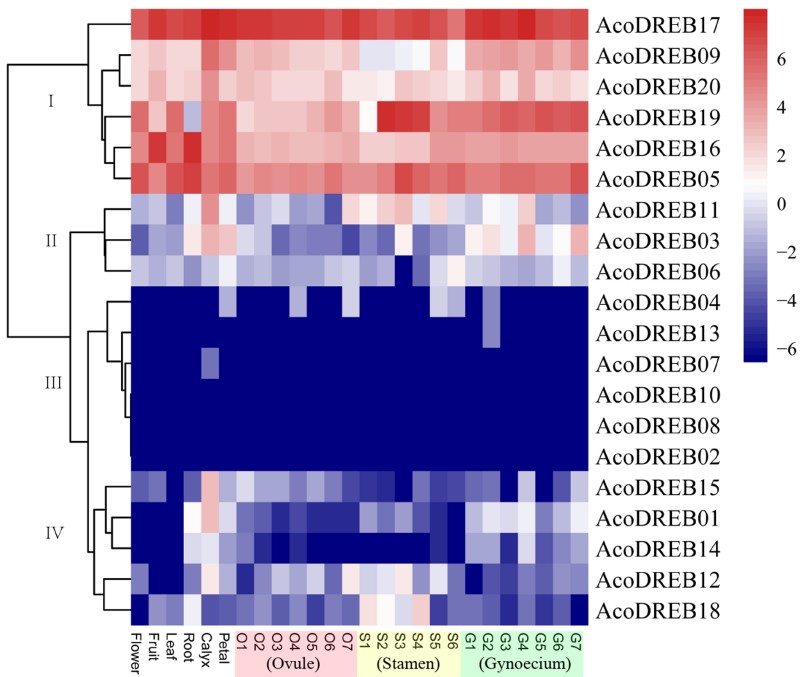

**Figure 7 Heatmap showing the expression levels of *AcoDREB* genes in different pineapple tissues.** RNA-Seq expression level can be understood using the givenscale and roman numbers on right-side shows clusters based on gene expression. O, S and G represent ovule, stamen and gynoecium, respectively.

Clustering analysis of the expression patterns of the 20 genes divided them into four clusters (Fig. 7). Of the six genes in cluster I, four (*AcoDREB05*, *16*, *17* and *20*) were highly expressed in all tissues, indicating that they may have important roles throughout plant growth. The expression level of *AcoDREB09* was lower in stamens than in other tissues, and *AcoDREB19* had the lowest expression in roots, suggesting that these particular cluster I genes may not be critical for the development of these respective tissues. The six genes in cluster III (*AcoDREB02*, *04*, *07*, *08*, *10* and *13*) had very low expression levels in all tissues, suggesting that these genes might only be expressed under specific conditions. Most of the genes in clusters II and IV had tissue- or stage-specific expression patterns. For example, *AcoDREB01* and *AcoDREB15* had higher expression in calyxes, suggesting that they may have a positive role in floral organ development. The higher expression of *AcoDREB06* in stage 6 stamens suggests a potential link to stamen maturity. *AcoDREB18* was highly expressed during stamen development. *AcoDREB11* was expressed in the ovule, stamen and gynoecium tissues, suggesting this gene may function widely during gametophyte development. *AcoDREB03* was highly expressed in the root, calyx, petal, and gynoecium.

## *AcoDREB* gene expression under abiotic stress

We analyzed *AcoDREB* gene expression under various abiotic stress conditions, including salt, drought, cold, and heat. Specifically, we examined the expression patterns of eight *AcoDREB* genes (*AcoDREB01*, *03*, *06*, *09*, *11*, *14*, *18* and *19*) in the MD2 variety of

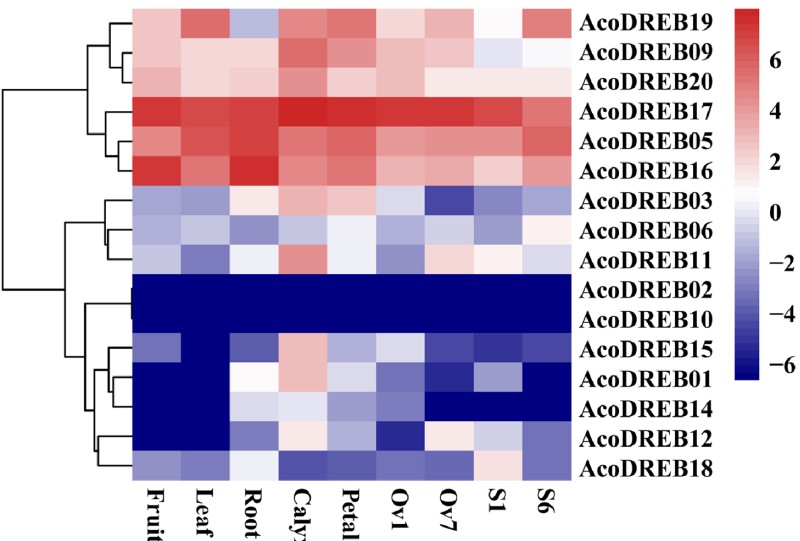

**Figure 8 The expression profiles of *AcoDREB* genes in nine tissues validated by qRT-PCR.** Validation of 16 genes at nine different tissues through qRT-PCR. Heat-map was constructed from relative gene expression in different tissues (qRT-PCR) data. 

pineapple using qRT-PCR with three biological and three technical replicates (Fig. 9; Table S2). Under all stress conditions, the relative transcript levels of the *AcoDREB* genes fluctuated during the 48-h analysis period.

We subjected pineapple plants to salt stress by treating them with 150 mM NaCl. The expression of all eight genes increased rapidly in the roots and reached a maximum level 12 h after the start of treatment. In shoots, five of the genes had highest expression levels at 12 h, and two genes had highest expression levels at 6 h. *AcoDREB06* expression in shoots decreased after salt treatment. The differential responses of the *AcoDREB* genes after NaCl treatment suggest that they have distinct roles in salt stress response (Figs. 9A–9H).

To analyze the response to drought stress, we treated plants with 350 mM mannitol. In the shoots, six genes (*AcoDREB01*, *03*, *11*, *14*, *18* and *19*) were down-regulated after 12 h. *AcoDREB09* was extremely sensitive to drought stress, and its expression level quickly reached a maximum at 6 h after treatment. Except for *AcoDREB06*, the expression levels of the analyzed genes did not change as much in the roots as they did in the shoots. Compared to the control plants, *AcoDREB03* and *AcoDREB11* were rapidly down-regulated in the roots. These expression pattern changes after mannitol treatment indicate the vital role played by *AcoDREB* genes in response to drought conditions (Figs. 9I–9P).

Cold stress drastically affects plant growth and development and causes major crop yield losses (*Cai et al., 2015*). The expression levels of the DREB genes were equally affected by cold treatment in the roots and in the shoots. In particular, three genes (*AcoDREB01*, *03* and *18*) responded rapidly to cold treatment, and their expression levels in the shoots peaked at 6 h. Two genes (*AcoDREB09* and *AcoDREB19*) reached their maximum expression levels in the shoots after 48 h (Figs. 9Q–9X).

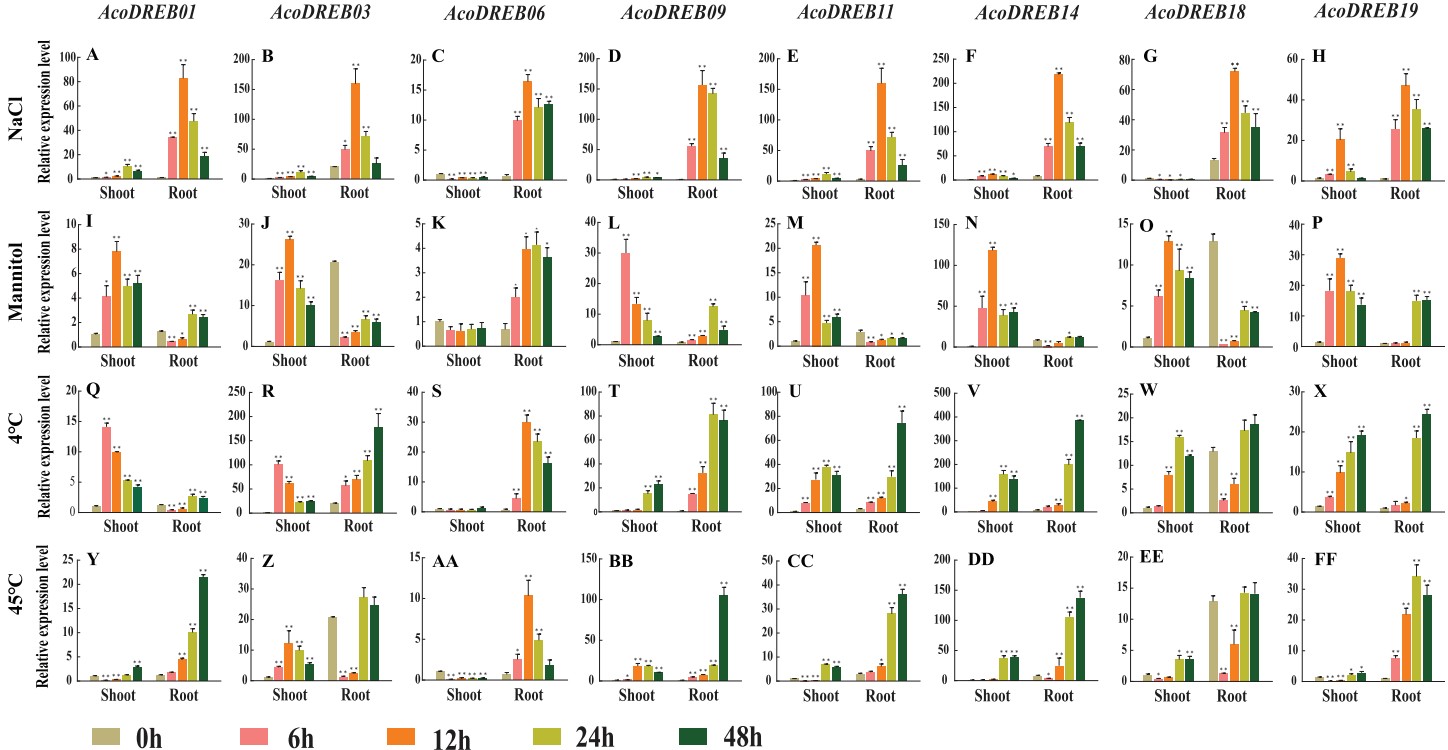

**Figure 9 qRT-PCR expression analysis of eight selected *AcoDREB* genes in response to different abiotic stress treatments.** (A)–(H) High salt (150 mM NaCl); (I)–(P) drought (350 Mm Mannitol); (Q)–(X) chilling, exposure to 4 °C; (Y)–(FF) high temperature, exposure to 45 °C. Mean expression value was calculated from three independent replicates. Error bars indicate the standard deviation. Data are presented as mean ± standard deviation (SD). Asterisks on top of the bars indicating statistically significant differences between the stress and counterpart controls (*$P < 0.05$, **$P < 0.01$). 

To analyze the effects of heat stress, the plants were subjected to 45 °C temperature. In the shoots, the majority of the analyzed genes were initially down-regulated then subsequently up-regulated. *AcoDREB03* was the only gene that was up-regulated in the shoots during the first 12 h. In the roots, the expression levels of four genes (*AcoDREB01*, *09*, *11* and *18*) gradually increased and peaked at 48 h. The expression levels of two genes (*AcoDREB03* and *AcoDREB14*) decreased rapidly after exposure to high temperature stress. Unlike the other genes, the expression of *AcoDREB06* in the roots peaked at 12 h. Collectively, these results indicate the involvement of *AcoDREB* genes in the response to heat stress in pineapple (Figs. 9Y–9FF).

## DISCUSSION

Climate change has drawn attention to the detrimental effects of environmental stress on plant growth and yield (*Chinnusamy, Schumaker & Zhu, 2004*; *Mittler, 2006*; *Suzuki et al., 2014*). Throughout their development, plants respond to stress by activating genes that induce a specific response to the stressor. These genes can be roughly divided into two categories. The first group includes functional genes directly responsible for the production of important stress resistance proteins, such as aquaporin, LEA protein and antioxidant

enzymes. The second group includes genes that encode regulatory proteins, such as TFs and protein kinases.

By recognizing and binding specific promoter *cis*-elements, TFs regulate the transcription of downstream genes. There are hundreds of TFs in higher plants, and they have important roles in plant reproductive development and physiological metabolism (*Liu, Zhang & Chen, 2001*). In response to environmental stress, TFs regulate plant growth and development by controlling a variety of downstream genes. The AtMYB4 TF protects plants from the harmful effects of UV radiation (*Hemm, Herrmann & Chapple, 2001*). Transgenic expression of *GmMYB22* in *Arabidopsis* enhances drought tolerance, salt tolerance, and ABA sensitivity (*Shan et al., 2012*). One class of bZIP proteins, the TGA/OBF family members, interact with NPR1 in the salicylic acid defense signaling pathway (*Singh, Foley & Onate-Sanchez, 2002*).

The DREB TFs contain a conserved AP2/EREBP domain, which is involved in the response to environmental stress. DREBs regulate genes that enhance plant stress tolerance by interacting with DRE *cis*-elements. In experiments with mutated DRE binding sites, DREB TF binding was abolished (*Dubouzet et al., 2003*; *Liu et al., 1998*). Other experiments dissected the preferential binding of DREB1A to two DRE sequences in *Arabidopsis* and *Oryza sativa* (*Dubouzet et al., 2003*; *Sakuma et al., 2002*).

Several studies have elucidated the functions and evolutionary history of *DREB* genes in many plant species including *Arabidopsis*, rice and maize. There have also been a growing number of studies that report the functions of *DREB* genes in stress response. *DREB* genes were first cloned in *Arabidopsis* in 1998 (*Liu et al., 1998*). *DREB1* and *DREB2* were involved in two separate signal transduction pathways that protect plants from low-temperature and dehydration conditions (*Liu et al., 1998*). In *Arabidopsis*, the expression of *VuDREB2a* from the legume cowpea (*Vigna unguiculata*) was found to enhance drought resistance (*Sadhukhan et al., 2014*). DREBs also protect plants from biotic and abiotic stress by regulating anthocyanin biosynthesis (*Song et al., 2019*). In addition, *MaDREB1–MaDREB4* (*Achr9G04630, Achr5G280, Achr6G32780* and *Achr11G24820*) are induced by ethylene in bananas (*Musa acuminata*) and regulate fruit ripening (*Kuang et al., 2017*). These examples from diverse plant species indicate that DREBs contribute significantly to plant growth and development.

Considering its high economic value, pineapple production would benefit tremendously from an improved understanding of the stress tolerance mechanisms in this species. We identified pineapple *DREB* genes and gathered the following information: the predicted pI and molecular weights of the encoded proteins, chromosome location, gene structure and motif, phylogenetic relationships, domain architecture, promoter *cis*-elements and expression profiles under abiotic stress.

We identified 20 *AcoDREB* genes (Table 1), which is fewer than the number of *DREB* genes in other monocots. For example, there are 57 *OsDREBs* (*Rashid et al., 2012*; *Nakano et al., 2006*) (*Oryza sativa*), 51 *ZmDREBs* (*Du et al., 2014*) (*Zea mays*), 52 *SbDREBs* (*Yan et al., 2013*) (*Sorghum bicolor*), and 27 *PeDREBs* (*Wu et al., 2015*) (*Phyllostachys edulis*). The lower number in pineapple suggests that some genes may have been lost during the evolution of this species. The predicted AcoDREB proteins ranged from

149 (*AcoDREB13*) to 463 (*AcoDREB20*) amino acids. The average length was 255 amino acids, which is very similar to that in rice and Chinese jujube (*Ziziphus jujube* Mill) (*Zhang & Li, 2018*). The predicted molecular weights (Mw) ranged from 16.32 (*AcoDREB13*) to 49.3 (*AcoDREB20*) kDa, and the predicted pI values ranged from 4.71 (*AcoDREB10*) to 9.68 (*AcoDREB07*) (Table 1). The ranges reported in other species include the following: 12.13–59.27 kDa and 4.6–10.64 pI in pepper (*Capsicum annuum* L.) (*Jin et al., 2018*) and 17.6–36.3 kDa and 4.5–11.07 pI in moso bamboo (*P. edulis*) (*Wu et al., 2015*). The predicted Mw and pI ranges in pineapple are roughly similar to those reported in other species, indicating some degree of conservation in the biochemistry and function of DREB TFs in plants. Therefore, based on previous studies of DREBs in other species, we can propose and test hypotheses about the characteristics and functions of DREBs.

To investigate the phylogenetic relationships of the *AcoDREB* gene family, we constructed an unrooted phylogenetic tree based on multiple sequence alignment of DREB amino acid sequences from pineapple, *Arabidopsis* and rice. The comparative analysis classified the *AcoDREB* genes into five subgroups (Fig. 3), and the numbers of genes in subgroups I to V were 3, 4, 4, 5 and 4, respectively (Fig. 3). In *Arabidopsis*, the *DREB* genes can be divided into six subgroups (A1–A6), with only one gene in the A3 subgroup. In the current study, *AcoDREB04*, *AT2G40220* (A3 subgroup) and *AT3G57600* (A2 subgroup) were on the same branch of the phylogenetic tree (Fig. 3), but we ultimately grouped *AcoDREB04* with the A2 subgroup based on sequence and domain analysis (*Nakano et al., 2006*). As a result, there were no *AcoDREB* genes that grouped together with the A3 subgroup. The genes of A3 subgroup may have been lost during the evolution of these species.

Analysis of the intron-exon structure of *AcoDREB* genes revealed a small number of introns. *AcoDREB05* had the highest number of introns (three), while many of the other genes lacked introns, which is consistent with previous reports in grape (*Vitis vinifera*) and jujube (*Zhao et al., 2014*; *Zhang & Li, 2018*). Some studies have proposed that introns could delay regulatory responses. To respond quickly to various stresses, genes must be rapidly activated. Having fewer introns would assist this process (*Jeffares, Penkett & Bahler, 2008*). In support of this hypothesis, we found a quick response to salt stress in the eight genes that we examined (Fig. 9).

The expression patterns of some *AcoDREB* genes resembled the expression patterns of homologs in other species. *AcoDREB19* was highly expressed in anthers (Figs. 7 and 8), which is similar to the expression of its homolog in rice (LOC_Os08g27220) (*Davidson et al., 2012*). Similarly, *AcoDREB16* and its homolog in rice (LOC_Os10g22600) are both highly expressed in roots. *OsDREB2A*, when overexpressed in rice, enhances salt stress tolerance (*Mallikarjuna et al., 2011*), without changing its total nutritional composition (*Cornwell, 2014*; *Cho et al., 2016*). Our analysis suggested that overexpression of some *AcoDREBs* in pineapple could help to develop new pineapple varieties with abiotic stress tolerance. Furthermore, we found that *AcoDREB05*, *16* and *17* displayed high expression levels in fruits (Fig. 7), indicating that they may play an important role in fruit

development. Therefore, it is possible that these genes may have applications in improving fruit quality through molecular breeding.

*DREB* genes respond to stress conditions through differential expression in shoots and roots (*Torres et al., 2019*). We therefore quantified the transcript levels of eight *AcoDREB* genes in pineapple seedlings subjected to different abiotic stress conditions. Under salt stress, eight of the DREB genes displayed similar expression patterns, and were induced in both shoots and roots (Figs. 9A–9H). Previous studies have reported that A1 subgroup members play important roles in the response to salt and drought stress in *Arabidopsis* (*Yamaguchi-Shinozaki & Shinozaki, 2006*). In our study, *AcoDREB01* and *AcoDREB03* from subgroup I were induced in plants subjected to salt and drought stress (Figs. 9A, 9B, 9I and 9J). Specifically, they were expressed in roots under salt stress, and in shoots under drought stress. These two genes also had similar expression patterns over the course of treatment, indicating that they may be coordinately regulated in response to salt and drought stress. Previous studies showed that *ScDREB10* was up-regulated after NaCl (150 mM) treatment and that its overexpression enhanced salt stress tolerance in *Arabidopsis* seedlings (*Li et al., 2019*; *Li et al., 2016*). We therefore infer that *AcoDREB01* and *AcoDREB03* may perform similar functions in pineapple.

Subgroup IV members *AcoDREB11* and *AcoDREB14* were both up-regulated under salt treatment and cold stress (Fig. 9). These expression changes are similar to those of the A5 subgroup member *GmDREB2* (*Chen et al., 2007*), suggesting functional conservation of these homologs in pineapple and soybean. At the same time, they also indicate functional conservation of the genes that belong to the same subgroup. Under various abiotic stresses, *AcoDREB06* expression decreased in the leaves and increased in the roots, indicating that enhanced expression of this gene could improve the resistance of roots to different abiotic stresses. On the other hand, the decreased expression of *AcoDREB06* in shoots suggests that it may also regulate other pathways that are critical to plant survival (Fig. 9). For instance, similar to the *Arabidopsis* gene *HARDY* (*AT2G36450*), it may improve drought and salt tolerance by reducing transpiration (*Abogadallah et al., 2011*). The RNA-Seq data indicated that *AcoDREB19* had very low expression in roots, but its expression increased significantly under different abiotic stresses.

The expression analysis for the eight selected genes were mostly in line with our expectations based on the predicted *cis*-elements in their promoters (Figs. 4 and 9). TC-rich and W-box elements were found in the promoters of *AcoDREB01, 06, 09, 11,* and *19*. Since these *cis*-elements have been identified upstream to genes that are key to plant defense in other species (*Laloi et al., 2004*; *Xu et al., 2010*), we speculate that these four genes play a similar role in resistance to pineapple diseases (*Hubert et al., 2014*; *Calderon-Arguedas et al., 2015*). These genes could potentially be used to breed disease-resistant pineapple seedlings.

## CONCLUSIONS

We identified 20 *AcoDREB* genes in pineapple, and collected information about their gene structures and expression profiles under various abiotic stresses. To the best of our knowledge, this is the first genome-wide analysis of *DREB* genes in pineapple. We have

shown that *AcoDREB* genes respond to a variety of abiotic stresses (drought, high salt, high- and low-temperature stress). Our findings provide preliminary data for further functional analysis of *AcoDREB* genes in pineapple, and information for developing new pineapple varieties with important agronomic traits such as stress tolerance.

## ACKNOWLEDGEMENTS

We would like to thank the reviewers for their helpful comments on the original manuscript.

### Funding

This work was supported by NSFC (U1605212, 31761130074, 31970333), a Guangxi Distinguished Experts Fellowship, and a Newton Advanced Fellowship (NA160391). The funders had no role in study design, data collection and analysis, decision to publish, or preparation of the manuscript.

### Grant Disclosures

The following grant information was disclosed by the authors:
NSFC: U1605212, 31761130074 and 31970333.
Guangxi Distinguished Experts Fellowship, and a Newton Advanced Fellowship: NA160391.

### Competing Interests

The authors declare that they have no competing interests.

### Author Contributions

- Mengnan Chai conceived and designed the experiments, prepared figures and/or tables, authored or reviewed drafts of the paper, and approved the final draft.
- Han Cheng conceived and designed the experiments, prepared figures and/or tables, and approved the final draft.
- Maokai Yan conceived and designed the experiments, analyzed the data, prepared figures and/or tables, authored or reviewed drafts of the paper, and approved the final draft.
- SVGN Priyadarshani performed the experiments, authored or reviewed drafts of the paper, and approved the final draft.
- Man Zhang conceived and designed the experiments, authored or reviewed drafts of the paper, and approved the final draft.
- Qing He performed the experiments, authored or reviewed drafts of the paper, and approved the final draft.
- Youmei Huang performed the experiments, prepared figures and/or tables, and approved the final draft.
- Fangqian Chen performed the experiments, prepared figures and/or tables, and approved the final draft.

- Liping Liu analyzed the data, authored or reviewed drafts of the paper, and approved the final draft.
- Xiaoyi Huang analyzed the data, authored or reviewed drafts of the paper, and approved the final draft.
- Linyi Lai performed the experiments, authored or reviewed drafts of the paper, and approved the final draft.
- Huihuang Chen analyzed the data, authored or reviewed drafts of the paper, and approved the final draft.
- Hanyang Cai conceived and designed the experiments, prepared figures and/or tables, authored or reviewed drafts of the paper, and approved the final draft.
- Yuan Qin conceived and designed the experiments, prepared figures and/or tables, authored or reviewed drafts of the paper, and approved the final draft.

### Data Availability

The Acomosus data is available at the Phytozome database:
https://genome.jgi.doe.gov/portal/Acomosus/Acomosus.info.html.

The data is also available at the Pineapple Genomics Database:
http://pineapple.angiosperms.org/pineapple/html/download.html.

### Supplemental Information

Supplemental information for this article can be found online at http://dx.doi.org/10.7717/peerj.9006#supplemental-information.

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
