# Peer review of "Identification and expression analysis of the DREB transcription factor family in pineapple (Ananas comosus (L.) Merr.)"

_PeerJ, doi:10.7717/peerj.9006_

## Round 0.1 · original submission · Major Revisions

After consideration of comments from all reviewers, I recommended major revision. Please address all the comments of the reviewers.

Reviewer 1 ·

Basic reporting

Literature references, sufficient field background/context provided.

Experimental design

Rigorous investigation performed to a high technical & ethical standard.

Validity of the findings

All underlying data have been provided; they are robust, statistically sound, & controlled.

Additional comments

Dear All
This paper was writen as good and detailed. It contributes to understanding of DREB genes in plants. Bioinformatics and wet-lab studies were used sucessfully in this study. Some minor problems were indicated as red in pdf file.

Annotated reviews are not available for download in order to protect the identity of reviewers who chose to remain anonymous.

Reviewer 2 ·

Basic reporting

no comment

Experimental design

no comment

Validity of the findings

no comment

Additional comments

Chai et al's manuscript entitled "Identification and expression analysis of DREB transcription factor family in pineapple (Ananas comosus (L.) Merr.)" first identifed 20 DREB genes in pineapple and characterized their gene's struture, tissure specific expression profile and expression profile under different stresses. This study provides new insight for future studies to find out the functions of DREB genes in pineapple. The manuscript is well written in English clearly. I recommend for publish in Peer J.

·

Basic reporting

This paper is focusing on identification and characterization of DREB genes in pineapple. Authors have used various in silico and in vivo techniques to achieve their goal. I would like to appreciated the effort put forward by the authors dissecting the role DREB family members in pineapple.
While appreciating the contribution made by the authors I would like to provide following suggestions to improve the manuscript.
1. There are considerable amount of formatting and sentence structure issues throughout the manuscript (Please see the attached annotated manuscript).
2. Most of the references were excluded through out the manuscript. Please back all the claims/facts you are pointing with valid references. Introduction and discussion need to be further strengthen while minimizing overlaps.
3. Figure titles needs to be revisited. Some suggestions are included in the annotated manuscript. Also, Fig 7. please include a colored text strip to indicate tissue/organ for O, S and G so readers can easily understand what tissue each abbreviation is referring to.Figure 4, - I would like to suggest a single line to indicate the promoter and use arrows to indicate the orientation of the cis-elements.
5. Gene conventions and formatting, abbreviation standards needs to be followed through out the manuscript. Please carefully revisit the manuscript.

Experimental design

The research presented here in the manuscript suit the aims and scope of the PeerJ.
1. Authors need to further elaborate the application of the research outcome in broader context.
2. Though the methods used in the study are appropriate, I noticed the following.
a. relevant references of tools are cited. There are deadlinks. Also it is important to include software versions in order to increase the reproducibility of the outcome.
b. Statistical analysis should be carried out for qPCR data in order to use the term "significantly" describe expression differences observed.
3. Consensus cis-element sequences and relevant references need to Incorporated either in methods or as a supplement.

Validity of the findings

Authors have generated comprehensive set of data in characterizing DREB family. However stress induced gene expression results need to be statistically assessed before make conclusions. There is a room to improve discussion incorporating and interpreting major advances made in DREB genes characterization in other plant spp. Conclusions need to address the objectives of the study and it is lacking in the current version of the manuscript.

·

Basic reporting

The article English text is clear but needs some improvements in the highlighted regions.
The introduction and background is sufficient to display the matter, but the references are not sufficient in the material and method section and also should be prepared based on the journal style.
The article structure is provided in a professional style and figures are sufficient to display your results, but the raw/supplementary data should be inserted to your paper for more justification.

Experimental design

The research was significantly within the Aims and Scope of the journal.
Research questions were well defined and expressed.
Methods were well described but some highlighted sections should be prepared in sufficient detail.

Validity of the findings

The description and conclusions sections can be improved with some unrepeated sentences during the manuscript.

Additional comments

Totally, the manuscript can be publish in the PeerJ, but after revisions; there are four general comments:
1. the paper text including headings and sub-headings should be prepared based on the journal template
2. the English text should be improved during the highlighted regions specifically in the materials and methods section
3. the supplementary tables and figures should be submitted
4. the references should be cited in the highlighted sections as well as should be prepared based on the journal style.

---

## Round 0.2 · Minor Revisions

The reviewers have made valuable comments on your manuscript for improving the quality of your manuscript, please take all comments into consideration for revision.

·

Basic reporting

While appreciating the authors for incorporating changes as requested, I like to point out that the manuscript is still require language improvement. This is essential for passing the significance of the study to targeted audience. I highly suggest authors to obtain help from a professional language editing service. That way authors will be able to improve the structure and flow of the manuscript.

Experimental design

Authors addressed most of my concerns regarding experimental design. However, things like how the qPCR data was analyzed is not given and relevant details are not specified in in the figure as well. I appreciate if authors can go through the manuscript again giving special consideration to methods in order to address deficiencies.

Validity of the findings

Authors have done fairly good job after the revision. However, impacts of the study can be further elaborated.

Additional comments

Please revisit figure legends (Fig 8 and 9). They need be further improved
"Figure.8: Heat-map for validation of DREB genes RNA-seq.
Validation of 16 genes at nine different tissues through qRT-PCR. Heat-map was constructed
from relative gene expression in different tissues (qRT-PCR) data." First part is on RNASeq while description is talking about qRT-PCR.

---

## Round 0.3 · Major Revisions

Dear Dr. Qin,

We have some final comments from the Section Editor, for you to consider for revision.

There is no significant reason provided why the DREB gene family is important for study in this species other than that it can be done.

Some of the attributes of this study may be the attempt to differentiate expression in five tissue types, and to track the expression profile to five different treatments. It is stated that RNA was extracted from five different tissues, yet the expression analysis was done only on downloaded data from six tissue studies from a database repository. It does not appear that any original expression data was actually reported.

Most of what is reported here is standard analysis protocols without an attempt to tie the information to a significant observation that may be important for the plant breeder. The report states that this is the first of its kind but does not really state what was learned from it except to say genes are expressed differently under different situations; how can the data provided be used? Some example or a pointer to a direct study would help.

As this study reflected on tissues and conditions it begs to have annotations added. Journal manuscripts are often scanned by text-mining software that locates and extracts core data elements, like gene function. Adding standard ontology terms, such as the Gene Ontology (GO, geneontology.org) or others from the OBO foundry (obofoundry.org) can enhance the recognition of your contribution and description. This will also make human curation of literature easier and more accurate. None of this was visible.

Reviewer 1 ·

Basic reporting

The article must be written in English and must use clear, unambiguous, technically correct text.

Experimental design

Original primary research within Aims and Scope of the journal.

Validity of the findings

Conclusions are well stated, linked to original research question & limited to supporting results.

·

Basic reporting

The authors has improved the writing style and structure of the manuscript.

Experimental design

Authors failed to include the statistical analysis they used to indicate significance of qPCR results (fig 9) in methods and figure legend is also incomplete even though it was requested under earlier review.

Validity of the findings

Section on qPCR is need to be improved to match with the comments made under experimental design.

Additional comments

No comments.

·

Basic reporting

The article is written in English clearly and contains technically correct text

Experimental design

The submission clearly defined the research question, which is relevant and meaningful.

Validity of the findings

All underlying data have been provided; they are robust, statistically sound, & controlled.

---

## Round 0.4 · Minor Revisions

I read your revised manuscript, and found some grammar errors, such as "line 305-306: We used transcriptome sequencing data and to analyze the...", please carefully check.

---

## Round 0.5 · Minor Revisions

The Section Editor has commented and said:

"The rebuttal was virtually non-existent with very few notable changes.

The tabs in Table S10 are not in English.

GeneIDs in table S8 would benefit from GO: terms in addition to assigned function. This was requested earlier, and would fit in nicely in this table.

I suggest that the authors be again asked to look at the previous request and provide an adequate rebuttal with amendments."

---

## Round 0.6 · accepted · Accept

I checked the revised version, change was made accordingly.